# The Regulatory Hierarchy Following Signal Integration by the CbrAB Two-Component System: Diversity of Responses and Functions

**DOI:** 10.3390/genes13020375

**Published:** 2022-02-18

**Authors:** Elizabet Monteagudo-Cascales, Eduardo Santero, Inés Canosa

**Affiliations:** 1Department of Environmental Protection, Estación Experimental del Zaidín, CSIC, 18008 Granada, Spain; elizabet.monteagudo@eez.csic.es; 2Departamento de Biología Molecular e Ingeniería Bioquímica, Universidad Pablo de Olavide, Centro Andaluz de Biología del Desarrollo, CSIC, Junta de Andalucía, 41013 Seville, Spain; esansan@upo.es

**Keywords:** CbrA–CbrB, signal transduction, carbon metabolism, two-component systems, *Pseudomonas*, *Azotobacter*, carbon catabolite repression

## Abstract

CbrAB is a two-component system, unique to bacteria of the family *Pseudomonaceae*, capable of integrating signals and involved in a multitude of physiological processes that allow bacterial adaptation to a wide variety of varying environmental conditions. This regulatory system provides a great metabolic versatility that results in excellent adaptability and metabolic optimization. The two-component system (TCS) CbrA–CbrB is on top of a hierarchical regulatory cascade and interacts with other regulatory systems at different levels, resulting in a robust output. Among the regulatory systems found at the same or lower levels of CbrAB are the NtrBC nitrogen availability adaptation system, the Crc/Hfq carbon catabolite repression cascade in *Pseudomonas*, or interactions with the GacSA TCS or alternative sigma ECF factor, such as SigX. The interplay between regulatory mechanisms controls a number of physiological processes that intervene in important aspects of bacterial adaptation and survival. These include the hierarchy in the use of carbon sources, virulence or resistance to antibiotics, stress response or definition of the bacterial lifestyle. The multiple actions of the CbrAB TCS result in an important competitive advantage.

## 1. Introduction

The remarkable metabolic versatility that is the hallmark of *Pseudomonas* is consistent with the number of genes in its genome that are intended to enhance its metabolic adaptation to drastic changes in environmental conditions [1,2,3]. In addition to the structural genes essential for providing the bacterium with basic functions, the efficient ability to adapt to environmental fluctuations lies in the presence of multiple regulatory systems capable of sensing external conditions and adjusting cell physiology accordingly [4,5].

Bacteria use the TCS as one of the main signal transduction mechanisms to respond to environmental signals. A prototypical TCS consists of a membrane-bound sensor histidine kinase (HK) which detects an environmental stimulus by a ligand-binding domain (LBD) and triggers a change in the phosphorylation state of a cytosolic response regulator (RR). The identification of the signal(s) that activate TCS is a crucial prerequisite to understand the corresponding regulatory circuit. In a canonical TCS, the sensor and effector domains are present on two separate proteins, allowing bacteria to respond to extracellular signals [6,7,8]. Other HKs are entirely cytosolic or possess cytosolic LBDs permitting responses to cytosolic stimuli. Typically, the sensor protein contains a variable N-terminal sensor domain linked to a conserved HK transmitter domain (also known as core domain), while the RR has an N-terminal receptor domain associated with an effector domain [9,10,11,12]. The signal transduction processes are typically initiated by the interaction of a signal molecule with the HK sensor domains, which alters the activity of the autophosphokinase domain and subsequent transphosphorylation of the RR partner. These signaling systems form a versatile modular mechanism with a positive (kinase activity) or negative (phosphatase activity) control over the activity of RR. In a positive control, autophosphorylation of a conserved histidine residue in the HK occurs in a reversible reaction, generating a high-energy intermediate, which in turn transfers the phosphoryl group to an aspartic residue of the RR [13,14,15]. On the other hand, in a negative control, dephosphorylation of the receptor domain of the cognate RR occurs as a consequence of the phosphatase activity of the HK, returning the system to its pre-stimulus state [16,17]. Therefore, a precise balance between RR phosphorylation and dephosphorylation processes is essential to generate an adjusted response [18,19]. The presence of a TCS has a dual impact in bacteria. Firstly, its maintenance requires an energetic expense, but it also provides an evolutionary advantage that enables them to sense environmental stimuli, which would not be viable in a one-component system (OCS), where the sensor domain is usually cytosolic. Nevertheless, the very large majority of signals that stimulate HKs is unknown and their identification represents a major research need in the field [20,21]. Additionally, analysis of the distribution of HKs and RRs of TCSs in bacteria has revealed that (1) the number of HKs and RRs increases slightly with genome size and (2) the amount of TCSs in bacterial genomes reflects the bacterial lifestyle [22,23]. Unlike the Archaean domain, where the number of HKs and RRs per strain is very different between species, especially in methanogens, bacterial genomes typically encode a similar number of both components [24].

The CbrAB TCS is involved in nutritional adaptation processes, and was first described in *Pseudomonas aeruginosa* as a regulatory system involved in the hierarchical utilization of various carbon sources [25]. To date, no orthologous systems have been described and the activating signal of the system has not been identified, although some authors suggest that it could be related to the C:N balance [25,26,27,28].

Transcriptome studies and ChIP-seq analysis have identified target genes directly activated by the RR CbrB, including the small regulatory RNAs CrcZ, CrcY and CrcX in different strains of *Pseudomonadaceae* [29,30,31]. These non-coding RNAs regulate the activity of the Crc/Hfq complex by direct binding, thereby interfering with the translation repression of the corresponding mRNA targets in the process of catabolite repression in *Pseudomonas* and *Azotobacter*, corresponding to a post-transcriptional regulation mechanism. In addition, cross-talk with NtrBC regulatory systems for the assimilation of certain amino acids as a carbon or nitrogen source has been reported [32,33,34,35], as well as possible interactions with extracytoplasmic function (ECF) sigma factors [36].

This review provides a broad perspective of the physiological and metabolic processes of *Pseudomonaceae* that are modulated by the CbrAB system, and their corresponding effects on adaptation and survival. For the first time, a detailed analysis is made of the direct effects of the regulatory system as a transcriptional activator, but also its role as a signal integrator through cross-communication with other central systems that have effects on many other physiological aspects such as virulence or catabolite repression. Furthermore, we describe the peculiarities in the structure and function of the HK CbrA and its mode of interaction with the activating signal, as well as inferring some evolutionary consequences that confer this functional particularity.

## 2. The Two-Component CbrAB System

The exclusivity of the TCS CbrAB in the family *Pseudomonadaceae* and its regulatory ranking is probably one of the reasons why several genera of this family show a remarkable versatility and adaptability to adverse conditions and sudden changes in environmental conditions. The *Pseudomonadaceae*, within the order *Pseudomonadales* and the class γ-proteobacteria, comprises fifteen bacterial genera grouped on the basis of 16S ribosomal RNA sequence similarity [37]. Within this family, the genera *Pseudomonas* and *Azotobacter* stand out for their environmental, clinical and biotechnological applications. Both of them are free-living bacteria and suited for the production of polymers of industrial importance, such as the polysaccharide alginate [38], the polyester poly-β-hydroxybutyrate [39,40], or polyhydroxyalkanoates [41]. Furthermore, many members of both genera are able to rapidly adapt to adverse conditions and possess an elevated redox capacity and tolerance towards organic solvents [42,43]. The relationship of the genus *Pseudomonas* with *A. vinelandii* is so close that some studies based on the phylogenetic similarity of some genes, sometimes more closely related between the two than between other members of the genus *Pseudomonas*, have led to the possible assignation of *Azotobacter* as a *Pseudomonas* [44]. It is therefore not surprising that both genera retain considerable similarities in terms of CbrAB-mediated carbon regulation, among other processes, leaving these genera as the exclusive hosts of this system, and exploiting the adaptive advantages it provides.

### 2.1. The Structural Peculiarity of CbrA and the Nature of Activating Signal of the CbrAB System

CbrA represents a new family of sensor HKs as its structure suggests it may link signaling to transport. Its modular structure includes a large N-terminal transmembrane (TM) region, also called the SLC5 (SoLute Carrier 5) domain, followed by a STAC (Solute Carrier 5—A Two-Component Signal Transduction-Associated Component) domain, a PAS domain and a C-terminal catalytic core consisting of a DHp (Dimerization and Histidine Phosphotransfer) domain and a CA (*Catalytic ATP-Binding*) domain (Figure 1).

CbrA is not a canonical HK as it is a fusion of a TM-domain-rich region forming a transporter with a segment comprising two potential sensor domains followed by the autokinase core domain. Although the number of TM regions may differ slightly between some *Pseudomonas* species (e.g., CbrA from *P. fluorescens* Pf-5 has 14 TM regions while *P. putida* KT2440 has 13 TM regions), the main difference with other HKs such as NtrB is the large TM region that keeps it anchored to the inner membrane. The TM region of CbrA shows structural analogy with the SLC5 transporter family, which belongs to the sodium symporter family (SSS). The SSS family include secondary transporters from all kingdoms of life that regulate the entry of diverse solutes (amino acids, sugars, ions and vitamins) across the membrane by means of the electrochemical sodium ion gradients [45]. Sodium ion gradients, in particular, are generated by primary sodium pumps [46]. Their relevance is evident in examples such as PutP from *Escherichia coli* and SiaT from *Proteus mirabilis*, which are Na^+^/proline and Na^+^/sialic acid transporters involved in bacteria–host interactions, or in *Helicobacter pylori*, where PutP-modulated proline accumulation is a key condition required for colonization of the human stomach [47].

It has been described in the literature that HKs and membrane transporters share similar but distinct functions in relation to nutrient uptake. While HK informs the cell of nutritional conditions in the medium, the latter transport various compounds into the cell. In this context, a non-canonical mechanism for TCS action has been reported, where activation is mediated by the transporters, and where the transport process stimulates the activity of an HK. The DctA transporter involved in the uptake of C4 dicarboxylates with the DcuSB TCS [48] or the PstSABC transporter system with the PhoRB TCS controlling the uptake of inorganic phosphate (Pi) [49] are good examples of such processes. In *E. coli*, the availability of Pi in the environment modulates the activation of PhoRB TCS through a signaling complex established between the PstSABC ABC-type transporter and the accessory protein PhoU [50]. PhoU interacts with the PstB protein of the PstSABC transport system and, under certain conditions, with the PAS domain of the HK PhoR, determining the balance between the kinase and phosphatase activity of the HK [51]. The conformational changes generated by Pi uptake cause PhoU to interact with PhoR, promoting its phosphatase activity towards its cognate RR PhoB. On the other hand, there are also cases in which a receptor, such as UhpC, senses glucose-6P, activates the HK UhpB and promotes the transcription of the hexose phosphate transporter, UhpT, through phosphorylation of the RR UhpA [52,53]. In eukaryotic organisms such as *Saccharomyces cerevisiae*, glucose receptors (such as Snf3p or Rgt2p) or amino acid receptors (such as Ssy1p) that stimulate the activity of HKs upon signal detection have also been described [54,55]. In all cases, fine coordination is critical to ensure that transporter expression is specifically induced by the nutritional conditions of the environment. Possibly, the convergence of both functions into a single protein is due to an evolutionary event (named gene fusion), resulting in a single protein in which transporter activity modulates the HK phosphorylation state [56,57,58].

Limited data are available on the mechanism of CbrA activation. Zhang et al. have shown that in *P. fluorescens*, CbrA not only senses histidine but is also able to internalize it through a signaling-dependent process, even when all its specific transporters (HutT_u_, HutT_h_ and HutXWV) are removed. Furthermore, they showed physical coupling between the SLC5 domain and the C-terminal segment that is required for activity, showing that the CbrA-induced histidine transport triggers a molecular stimulus that activates the CbrAB system [59]. In *P. putida*, it has also recently been reported that L-histidine transport through the SLC5 domain in CbrA is not coupled to an electrochemical Na^+^ gradient but probably to an H^+^ gradient [60]. This effect might be due to the lack of conservation in CbrA of certain amino acids of the Na^+^-binding SLC5 domain (Ser340 and Thr341) of the Na^+^/proline symporter PutP [61,62].

However, it is still unclear whether the presence of an SSS domain in certain HKs is a prerequisite for signal perception or whether it merely functions as a transporter. In an attempt to answer that question, Monteagudo-Cascales et al. designed truncated proteins that lack the two putative CbrA sensor domains (SLC5 and PAS domain) and analyzed kinase activity indirectly through measuring *PP2810* transcript levels, a CbrB target that, unlike *crcZ* and *crcY*, is very sensitive to activity changes in the Cbr system [63]. The study showed that overexpression of a truncated CbrA variant lacking the SLC5 domain, resulted in 30% activation of *PP2810* as compared to the wild-type protein. This suggests that SLC5 domain activity may modulate CbrA autokinase activity, although it cannot be excluded that other factors such as alterations of subcellular localization may be contributing to this reduction [63]. On the other hand, a mutant containing an in-frame deletion of the PAS domain was unable to activate the *PP2810* target at any differential protein levels in any condition, suggesting that it is involved in the perception of an activating signal of the CbrAB system. To validate this hypothesis, further analysis of ligand–PAS domain interactions would be needed to identify this signal.

In this line, Wirtz et al. further showed partial in vitro autokinase activity of an equivalent variant of a ΔSLC5 CbrA, which was able to transphosphorylate CbrB, independently of L-histidine transport [60]. They also showed interaction of the transmembrane SLC5 domain with L-histidine that occurred independently of the phosphorylation of the DHp domain. L-histidine did not alter the autokinase or transphosphorylation activity of CbrA, suggesting that a yet to be identified intracellular metabolite is perceived by CbrA as a signal. The putative interaction of the PAS domain with histidine yielded a discrete increase in the Tm value (0.73 ± 0.13 °C) measured with Nano differential scanning fluorimetry (NanoDSF), which, however, remain to be validated by direct ligand binding measurements such as calorimetry or Biacore. To test signal transduction in vivo, Wirtz et al. assayed *crcZ* activation in the ΔSLC5 and ΔPAS-CbrA truncated versions, and detected, as above, partial induction in the ΔSLC5 background, and little impact in the ΔPAS-CbrA variant. Considering that *crcZ* expression has a high basal level and is rapidly induced at low doses of CbrB~P [63], it would be more appropriate to define the gradual activation of the system, and the relevance of individual CbrA domains for signal transduction, on a more sensitive target, such as *PP2810*.

In either case, both experimental approaches reveal that the SLC5 domain is not involved in the transport or detection of the possible activating signal, which has yet to be identified. The data suggest that the inducing signal(s) are intracellular signals or are internalized independently of CbrAB.

Furthermore, the construction of a chimera comprising the PAS-HK domain of CbrA and the 13 TM regions of the SLC5 domain of the CrbS has demonstrated that substrate transport is not necessary for signal transduction [64]. The CrbS/R system regulates acetate utilization in *Vibrio cholerae*, *P. aeruginosa* and *P. entomophila* [65,66]. Sepulveda et al. showed that the corresponding chimera recovered the ability to grow on histidine compared to a Δ*cbrA*Δ*crbS* double mutant, thus demonstrating the catalytic activity of the PAS-HK domain of CbrA [64]. However, the efficiency of the chimera in mediating transcriptional activation of a CbrAB target gene has not been studied and it is difficult to interpret the level of activation obtained. Monteagudo-Cascales et al. demonstrated in *P. putida* that low activity of a truncated version of CbrA was able to restore its ability to use histidine as a source of carbon, although the activity of *PP2810* was undetectable. This indicates that the use of histidine in this case is less sensitive to low levels of CbrAB activity, whereas activity as expression of its targets requires higher levels of CbrB~P [63]. Therefore, although the efficiency of activation of the chimeras remains to be verified, these studies provide evidence for the detection of a signal by the PAS-HK domain. The fusion of a SLC5 transporter with an autokinase domain in signal transduction proteins such as CbrA or CrbS appears to play an important role in the detection of a signal, which triggers the response when there is sufficient active protein in the cell. It seems more plausible that the signal is detected intracellularly by the PAS domain through interaction with a metabolite, or as a function of a ratio between several, as occurs in other HKs that detect C:N balances [67,68]. Previous studies of the metabolomic analysis of CbrAB in *P. aeruginosa* already support this hypothesis [26]. However, this hypothesis does not rule out a modulatory or other additional role of the SLC5 domain in signal detection.

Although CbrA was initially associated with the group of HKs with sensing mechanisms in the TM region, several premises lead us to hypothesize the evolutionary origin of CbrA. One of them is the narrow phylogenetic distribution of the CbrAB system in the *Pseudomonadaceae* family as mentioned above. However, perhaps the most interesting is the peculiar structural organization of CbrA, which may result from the fusion of a transporter and an HK where histidine transport could stimulate the HK through a conformational change of the SLC5 domain.

### 2.2. CbrB as a Transcriptional Activator of σ^N^-Dependent Promoters

CbrB belongs to the third largest group of DNA-binding RRs, the NtrC subfamily, constituted by transcriptional activators of σ^N^ factor-dependent promoter. This class of RRs oligomerizes typically when phosphorylated, to higher-order forms (hexamers, heptamers, etc.), acquiring ring structures that contact with the RNA polymerase holoenzyme while forming a closed complex with the promoter and induce its isomerization into a transcriptionally proficient open complex [69,70]. As most members of this subfamily, CbrB is composed of three domains: the REC receiver domain, the AAA^+^ ATPase domain and the HTH (*Helix-Turn-Helix*) DNA-binding domain (Figure 1). The REC domain is located at its N-terminal end and is the receptor domain of the signal detected by CbrA, who determines the specificity of the response regulator. It contains the conserved aspartic residue (Asp-52) susceptible to phosphorylation by CbrA and exerts a regulatory role over the core domain often through control of oligomerization [31,69,70,71]. The AAA^+^ ATPase domain is the central domain of CbrB involved in the interaction with the σ^N^ factor of RNA polymerase and comprises the Walker A (GESGTGKE) and Walker B (ADGGTLFLDE) motifs for ATP binding and hydrolysis. Finally, the HTH domain is the DNA-binding domain and is located at the C-terminal end (Figure 1). However, CbrB is a rather peculiar σ^N^-dependent activator, since, unlike NtrC, it is still able to activate transcription of its target genes in vitro by 40% in its unphosphorylated and theoretically inactive form [72]. The two components of the CbrAB system show sequence similarity to the NtrBC family members. For *P. aeruginosa* PAO1, the CbrA C-terminal region shares an amino acid sequence identity of 34% with NtrB [25] of *E. coli*, and CbrB and NtrC display a 43% identity in a region of 387 residues to the NtrC [72].

## 3. Genomic Organization and Expression of the CbrAB TCS

The genes *cbrA* and *cbrB* are clustered in the same transcriptional unit according to the Database for prOkaryotic OpeRons (DOOR) operon prediction software [73]. The expression of *cbrAB* is directed by the P*_cbrA_* promoter located upstream of the *cbrA* coding region. In addition, an internal promoter, named P*_cbrB_*, has been identified by expression in *P. putida* and *P. aeruginosa* at the 3′ end of *cbrA*, which also promotes transcription of *cbrB* [25,27] (Figure 2). Our own unpublished results indicate that *cbrB* is primarily expressed from P*_cbrB_*, although there is a small basal expression, considered not significant, from P*_cbrA_*.

Downstream and in a separate transcriptional unit to *cbrAB* is the *crcZ* gene, which encodes a regulatory RNA. CrcZ, together with other similar RNAs (CrcY in *P. putida* KT2440 and A. *vinelandii* AEIV, and CrcY and CrcX in *P. syringae* pv. *tomato* DC3000), exerts an antagonistic effect on the catabolic repression process mediated by the Crc/Hfq complex in bacteria of the *Pseudomonaceae* family [30,72,74,75,76]. Although some RNAs are genetically redundant, Liu et al. proposed through a mathematical modelling that the *crcZ* and *crcY* co-existence in *P. fluorescens* SBW25 may be related to different associated biological functions [77]. In *P. putida*, the P*_crcZ_* and P*_crcY_* promoters (and P*_crcZ_* in *P. areruginosa*) are dependent on the alternative factor σ^N^ and are directly activated by CbrB. A read-through transcription of *crcZ* from P*_cbrB_* allows maintaining high basal levels of CrcZ to control Crc/Hfq availability under carbon catabolite repression (CCR) conditions [72,78].

The genomic organization of *cbrA*, *cbrB* and *crcZ* is highly conserved in different species of the genera *Pseudomonas* and *Azotobacter* of the family *Pseudomonadaceae*, the only genera where it has been described (Figure 2). The conservation of synteny makes the Cbr system a TCS unique to the *Pseudomonadaceae*, and unravelling the mechanisms of regulation, detection and transduction of the activating signal by the histidine kinase CbrA is of great scientific interest.

Detailed analysis of the *cbrA* promoter region revealed the presence of a small open reading frame, which is partially overlapping the translation start of CbrA. The role of this gene, named *cbrX,* has been investigated in *P. putida* [63]. The sequence of *cbrX* encodes a small peptide of only 58 amino acids of currently unknown function, and it is co-transcribed in the same transcriptional unit as *cbrA*. *cbrX* overlaps 17 nt with the sequence of *cbrA*, but both genes are translated in different open reading frames. In addition, the mRNA contains a possible semi-conserved sequence Shine–Dalgarno (TCGAGG) located 5 bp upstream of the first ATG, a possible translation initiator of *cbrX*, which could constitute the ribosome binding site for CbrX translation. Secondary structure prediction of CbrX and homology modelling revealed the presence of two α-helices that could be embedded in the inner membrane of the cell, close to the SLC5 domain of CbrA (Figure 2C).

Sequence comparison of CbrX in the databases showed a high degree of similarity with MSF (*Major Facilitator Superfamily*) transport proteins that are primarily involved in the uptake of nutrients or the efflux of toxic compounds through the membrane. Thus, the high degree of conservation of *cbrX* within the *Pseudomonadaceae* family and its structural analogy to MSF-type transport proteins suggest that CbrX could participate in the process of reception/transduction of the activating signal of the system or exert a regulatory mechanism on the expression of *cbrA*. Mutagenesis studies on the *cbrX* sequence and its effect on *cbrA* expression revealed the existence of a translational coupling between the two, where full translation of the *cbrX* reading frame is required for *cbrA* translation. However, although mutations in the *cbrX* sequence reduce *cbrA* expression, Cbr-mediated activation is not altered, suggesting that basal levels of phosphorylated CbrA are sufficient to phosphorylate CbrB, and thus promote efficient activation of at least those target genes that require little active CbrB, such as *crcZ* [63].

## 4. Transcriptomic Analyses for the CbrB Regulon Determination

One of the most challenging issues in understanding the regulation by CbrAB and the adaptive advantages it may provide to the bacteria is the identification of the genes directly activated by CbrB. However, given the epistasis caused by the regulator being on top of the regulatory hierarchy, it is often difficult to establish the precise regulatory cascade. In particular, the activation of the CbrAB system has an effect on the post-transcriptional regulation of the Crc/Hfq complex, since its activity is directly controlled (in most cases) by the presence of the CrcZ/CrcY/CrcX regulatory RNAs that bind directly the protein complex, causing its inactivation. A global high-throughput approach to discriminate between the actions of CbrAB of Crc/Hfq regulation has the limitation that the induction conditions of the two systems are reversed (limitation versus excess carbon availability) and the interpretation of the results is complex. Although several experimental approaches have been performed with this aim, it remains to be defined whether some genes are directly activated by CbrAB or repressed by Crc/Hfq and its transcription induced by the antagonistic CrcZ/CrcY [29,79,80]. This leaves gaps in the regulatory map to be filled in.

Another limitation that hinders the precise definition of the CbrB regulon is based on the fact that other regulatory elements often concur for a complete induction of some of the controlled pathways, such as the assimilation of some amino acids that require a specific induction, and the participation of specific regulators such as ArgR or HutC [81,82,83,84,85]. Therefore, the conditions must contain a set of inducers that can both constitute a carbon source and increase carbon availability, which limits the induction of the system [33]. Transcriptomic analysis have been performed on a *cbrB*^−^ background combined with others such as *crcZ*^−^ (which is Crc-constitutive in *P. aeruginosa*), or *ntrB*^−^ or *cbrBntrC*^−^ backgrounds to decipher the tangled network of transcriptional activation by different regulators [29,33,86].

To overcome these restraints and define the direct binding of CbrB to the promoter regions of target genes, it seems appropriate to perform in vivo binding analyses under conditions of committed activation. Apart from the previously described targets for CbrB in *P. putida* the small RNAs CrcZ and CrcY [74], the ChIP-seq analysis led to the identification of several undescribed targets including the putative efflux pump encoded by the operon *PP2810*-*PP2813*, the porin OprD, PP3074 encoding a putative permease or the histidine kinase PP3420 [31]. Other targets that have also been positively controlled by CbrB are CrcZ in *P. aeruginosa* and *A. vinelandii* [75,76], CrcZ and CrcX in *P. syringae* [30], the *lipA* (lipase) gene in *P. alcaligenes* [87,88], and the *hutU* (histidine utilization) operon of *P. aeruginosa* and *P. fluorescens* SBW25 [35,89,90].

Several approaches have been made to define the CbrB binding sites in *P. aeruginosa* and *P. alcaligenes*, to determine a consensus CbrB binding sequence in the promoters of *crcZ*, *lipA* or *hutU* [86,87,89]. The proposed model contained two non-palindromic subsites with variable spacing (3 to 12 nucleotides between them) [86]. To explore this aspect further, Barroso et al. conducted an intensive mutagenesis study on the possible binding sites of three promoters in *P. putida* (*crcZ*, *crcY* and *PP2810*), and included a possible third subsite in the analysis. The subsites were named F1, R1 and R2 and the first one was in direct orientation and R1 and R2 in reverse orientation (Figure 3) [31]. The purpose of the work was not only to define a consensus sequence for the binding of CbrB, but also to investigate the involvement and relevance of each of the subsites in the activation of gene transcription. Substitution of each one of the subsites independently for each target affected the transcriptional activation in a different manner. Although all substitutions had a substantial effect on *crcZ* expression it was the substitution of R2 the one showing the most drastic effect for *crcZ* transcription. Nevertheless, for *crcY* it was the substitution of F1 the one that reduced the activity most, and substitution of R1 for *PP2810* (represented as grey boxes in Figure 3). These data indicate that there is no preferential binding subsite that implies cooperativity in CbrB binding and reveal the different relevance of each depending on the context, suggesting that CbrB can bind to DNA in a relatively relaxed (variable) manner. Nevertheless, although the sequence conservation, seems to be only required at two of the three sites, the spacing between them is rather conserved. On the other hand, the involvement of IHF may contribute to the contextualization to favor the activation of CbrB at sites with different affinities.

The alignment of the binding sites of the experimentally tested regions to which CbrB binds, allowed the definition of a consensus sequence and establish the putative definition of CbrB binding sites (TGTTAC-N_12-14_-GTAACA-N_15–19_-GTAACA; Figure 3A) [31]. This pattern was consistent with that previously described for *P. aeruginosa*, although the latter only included subsites F1 and R1 [86], so in this review, and with the updated information, we have incorporated subsite R2 to complement the sequence (Figure 3). The work of Abdou et al. (2011) obtained evidence both in vivo from mutational analysis of the *crcZ* promoter, and in vitro from EMSA analysis, proposed two putative CbrB subsites for the *hutU* promoter of *P. aeruginosa* and *lipA* regions in *P. aeruginosa* and *P. alcaligenes* [86]. Nevertheless, although the consensus was close to the ones proposed here, the spacing between them seems insufficient to accommodate CbrB. Visual detection of other sequences with some degree of conservation suggests that the model could be re-evaluated. We present a LOGO sequence with the reevaluated CbrB-binding sites of the experimentally assayed promoters (Figure 3B).

CbrB, as a σ^N^-dependent transcriptional activators that usually bind DNA as dimers or higher-order complexes to palindromic sequences located far upstream of the promoter, usually need the assistance of DNA bending proteins, such as IHF, to activate RNA-polymerase transcription at a distance via DNA loop formation [91,92]. However, the CbrB binding sequences identified were not palindromic, and any attempt to identify CbrB multimerization complexes in solution were unfruitful, neither in the presence of DNA fragments containing the binding sequences, nor with prior in vitro phosphorylation of the protein [30]. The absence of palindromic sites and no perceptible dimerization in vitro for CbrB suggest that it does not bind as a dimer, but it appears to initially bind as a monomer to each one of the subsites. Although there is no evidence of oligomerization of CbrB in solution, the formation of structures of higher order is the most rational prediction to stabilize the complex in vivo. It appears that each particular binding site has an accumulative effect upon CbrB binding, although none of the three is essential for transcriptional activation [31].

Overall, it appears that CbrB-mediated activation is multifactorial and requires the convergence of several elements such as the conservation of the sequence in three binding subsites, the appropriate spacing between them, the intervention of an IHF-type element that favors appropriate DNA curvature, or the presence of a conserved σ^N^ sequence that controls the affinity of the RNA polymerase. Additionally, the need for specific induction by each specific regulator or the interaction of other global control systems will also determine the activation of a certain pathways that will drive cellular physiology and adaptation under certain environmental conditions. An example of a metabolic pathway that brings together all these elements is the histidine assimilation pathway in *Pseudomonas*, which is under the control of the global CbrAB and NtrBC systems, and is also repressed by the specific regulator HutC under conditions of specific induction by histidine or the intermediate urocanate [85,93]. The molecular mechanism of induction of this pathway will be discussed in detail in the following section.

## 5. The CbrAB-Mediated Control of the Amino Acids Catabolism

Initial studies of the TCS CbrAB showed that mutants affected in the components of the system were unable to assimilate certain amino acids as a carbon, or carbon or nitrogen sources. In fact, the system was identified as the result of mutagenesis studies to identify deficiencies in arginine utilization [25,94]. Several mutants in *cbrA* or *cbrB* in *Pseudomonas* such as *P. putida* KT2440 [26,27,29,33,63,72]*, P.*
*aeruginosa* PAO1 [25,26,28,32,95] and *P. fluorescens* SBW25 [28,84] have been described to be uncoupled in the assimilation of different amino acids. In *A. vinelandii,* which is unable to grow using amino acids as the sole carbon source itself [96], a CbrB mutant is uncoupled on glucose uptake [97].

In addition, mutants in *cbrA* or *cbrB* are impaired in the utilization of other non-preferential carbon sources. In *Pseudomonas*, succinate is a preferential source, and glucose is a less assimilable carbon source, since members of this genus do not have a functional glycolytic pathway, but instead its assimilation relies on the Entner–Doudoroff pathway (EDP) [98]. This is also the case for *A. vinelandii* [99]. In these strains, the carbon sources most easily assimilated repress assimilatory processes of other carbon sources. As a result, these latter sources are consumed once the former have been depleted, and such regulatory processes are referred to as catabolite repression. For different microorganisms, the repression capacity of carbon source or amino acids may vary. In the presence of less-preferred substrates, the activity of the CbrAB TCS is highly induced. For instance, in *P. putida*, the order of compounds in the growth medium that guarantee decreasing repression are oxaloacetate (OAA), arginine, alanine, pyruvate, lactate, histidine, or proline, while in *P. aeruginosa*, this order is different and was established to be OAA, histidine, pyruvate, alanine or arginine [26]. Succinate constitutes the most potent repressor source of the CbrAB in both strains, reaching almost levels observed in the rich LB medium [26]. Furthermore, the sequential hierarchy of amino acid utilization in *P. putida* in a rich medium has been reported, establishing which amino acids are assimilated first and which metabolic processes are favored, assigning a key role to Crc/Hfq in this process [100,101,102].

Several studies have focused on the implication of the CbrAB TCS in the uptake of different amino acids, and have determined what metabolic pathways are active under carbon-limited conditions [25,28,33]. The processes in which the direct involvement of CbrAB has been described are outlined below.

### 5.1. Arginine Catabolism

Arginine assimilation in *Pseudomonas aeruginosa,* involving multiple catabolic pathways, is a central example to illustrate the metabolic versatility of this organism [103]. Under aerobic conditions, the main pathway for arginine uptake is the arginine succinyltransferase (AST) pathway, where arginine is transported through the ABC transporter system *aotJQMP* [104] and converted into glutamate through the arginine succinyltransferase encoded by the *aruCFGDBE* operon [82] and the *gdhB* gene [105]. On the other hand, it is quite common to find a specific regulation superimposed on the metabolism of many amino acids, which is controlled by a specific regulator that activates their assimilation when the amino acid is present. In the case of arginine, the specific regulator ArgR is essential for the induction of the necessary genes [81]. Moreover, CbrAB has been shown to participate in the activation of the *aot* operon [25] in concert with ArgR [104], but is dispensable for *aruC* activation, which is only dependent on ArgR [25].

The other arginine assimilation pathways in *P. aeruginosa* are the arginine deiminase (ADI) pathway encoded by the *arc* operon [106], which provides ATP to support slow growth under anaerobic conditions, the arginine decarboxylase (ADC) pathway to supply putrescine when arginine is abundant, and the arginine dehydrogenase (ADH) pathway. The latter was considered the second arginine utilization pathway in *P. aeruginosa* under aerobic conditions, as it allowed residual growth in mutants of the AST pathway, but did not allow Arg utilization under aerobic conditions when both pathways were interrupted [103]. In addition to *P. aeruginosa*, arginine utilization has been studied in several different *Pseudomonas* species [107]. *P.*
*putida* contains, in addition to the AST pathway, the arginine decarboxylase (ADC), arginine deiminase (ADI) and arginine oxidase catabolic pathways [108]. The first two pathways are also present in *P. fluorescens* and *P. mendocina* [107]. Although mutants in these genes are known to be impaired in the use of arginine, the direct involvement of CbrAB in the regulation of the process has not been described to date.

### 5.2. Histidine Catabolism

Histidine assimilation in *Pseudomonas* is a good example of the interplay of multiple regulatory systems in the catabolism of an amino acid that can be used as a source of carbon, nitrogen, or both. The histidine catabolism pathway is highly conserved among bacteria and involves the removal of ammonia from histidine to yield urocanate, hydration of urocanate, resulting in imidazolone propionate (IP), and cleavage of the IP ring to yield formiminoglutamate (FIG). In *Pseudomonas*, the imino group of FIG is hydrolyzed to yield ammonia and formylglutamate (FG), which is then hydrolyzed to give formate and glutamate (see [109] and references therein for a full review). The five activities involved in this process are encoded by *hutH*, *hutU*, *hutI*, *hutF* and *hutG*, respectively, which are grouped in two transcriptional units directed by P*_hutU_* and P*_hutF_* promoters (Figure 4). In this reaction, one molecule of histidine yields three metabolizable nitrogen atoms, triplicating the yield of the ammonia breakdown. Histidase and urocanase, which are the first and second enzymes in the pathway, are inducible by urocanate, although histidine can also induce this operon by serving as a precursor [110]. The substrate-specific induction of the *h*istidine *ut*ilization *hut* genes in *Pseudomonas* is negatively regulated by the HutC repressor that binds to the *hutC* promoter in the absence of urocanate [84,85]. In contrast to the *hut* promoters of *P. putida* in the analogous operon in *P. fluorescens*, strain SBW25 is more complex, since it contains an insertion of five genes between *hut**U* and *hut**H* encoding a high-affinity transporter for urocanate (*hutT_u_*), an additional high-affinity ABC-type transporter (*hutXWV*) and a gene coding for an histidase (*hutH1*), which, however, is not required for growth on histidine [35,84]. This last module is also present in the genome of *P. aeruginosa*, which bears an additional ABC transporter system encoded by *orfXYZ*, and an *orfT* encoding another histidine transporter (Figure 4B).

The control of the *hut* pathway is a central model for the study of gene regulation in several families. In particular, in pseudomonads, it coordinates processes of carbon and nitrogen metabolism and catabolic repression, as well as temperature-dependent regulation [109]. The regulators involved in these processes are HutC, CbrB, and NtrC. All three bind to the intergenic region between *hutCD* and *hutU*, at the operator sites of *hut* promoters that share binding sequences, resulting in steric hindrance in the regulatory mechanism.

When histidine is used as an N source, NtrC activates the transcription of *hut* genes, and autoactivates its own expression, in analogy to many σ^N^ activators [16], but overexpression will produce excess ammonium, leading to NtrBC inactivation. The specific histidine-repressor HutC then represses *ntrBC* expression, resulting in an additional negative feedback loop. NtrC binds two palindromic subsites upstream the *hutU* transcriptional start site, at a considerable distance from the +1 position, which is a typical feature of canonical activators of σ^N^ promoters. Under these conditions, CbrB cannot bind to the promoter region because its binding site overlaps with one of the NtrC binding subsites (see Figure 4). Intriguingly, HutC also binds the *ntrB* promoter, and transcriptomic analyses of a HutC mutant growing on succinate and histidine showed upregulation of 46 genes involved in nitrogen regulation, but also of other genes that are related to biofilm formation, motility and virulence, suggesting a regulatory role beyond histidine catabolism [85].

Although the interplay between the regulation by HutC and NtrBC has been demonstrated, there is no evidence for cross-talk between HutC and the CbrAB system, although both regulatory elements come into play when histidine is used as carbon source. Under histidine-limiting conditions, CbrB binds to the *hutU* promoter region and activates its transcription, facilitating a basal activation for histidine catabolism and preventing its accumulation [84] (Figure 4C). HutC would act as a control element to direct the alternative activation of the CbrAB and NtrBC systems to drive the correct homeostasis in the carbon-nitrogen balance. Additionally, the high levels of the sRNAs CrcZ (and CrcY), which are highly expressed by CbrB activation, alleviate catabolite repression by Crc/Hfq, shown to repress mRNA translation of the *hut* transcripts themselves, resulting in an additional activation of histidine catabolism under these conditions [35].

On the other hand, although the activating signal of the CbrAB system has not been identified, it has been reported that CbrA is able to transport histidine in *P. fluorescens* through the SLC5 domain [59,60], and thus may provide a specific signal for the activation of the *hut* genes. It is likely that CbrAB detects histidine limitation and triggers HutC activation and consequent repression of the NtrBC system, which is active at higher histidine concentrations. Nevertheless, the functional link between histidine and urocanate transport and potential CbrAB activation is not well established. A variety of alternative permeases exist that may transport both compounds, as well as others that are occasionally poorly annotated (such as ProY, proline transporter), sometimes within the *hut* operon have been identified. Mutagenesis analysis has made it possible to assign a function to some of these transporters and in this way HutT has been identified as the main histidine transporter in *P. putida*. HutT is encoded by a single gene (*hutT*) which is located within the same transcriptional unit as *hutU*, and thus, possibly subject to the same regulation [111]. However, in *P. fluorescens* SBW25, there are two *hutT* orthologues encoding a putative histidine transporter, located within the *hut* genes; *hutT*_h_ and *hutT*_u_. The first one seems to be involved in histidine uptake, while the later seems to be a urocanate transporter [112]. In addition, the putative histidine transport system of the ABC type (*hutXWV*) also participates in histidine uptake [59]. Surprisingly, a *P. fluorescens* SBW25 mutant, devoid of all these histidine uptake systems, was still able to grow with histidine [112], illustrating the complexity of the histidine uptake system in this organism.

### 5.3. Proline Catabolism

*CbrB* mutants in *P. aeruginosa* and *P. putida* are impaired in the utilization of proline as a carbon source [25,33] and have growth defects in this carbon source. Proline utilization is achieved via a single catabolic pathway [113], where the first step requires the entry of this amino acid into the cells through the specific PutP transporter, which is coupled to the entry of sodium ions [114,115]. Proline is converted into glutamate by the bifunctional PutA enzyme with proline dehydrogenase (PDH) and pyrroline-5-carboxylate dehydrogenase (P5CDH) activities. This pathway is well conserved in enterobacteria and *Pseudomonas* [116,117,118]. In *P. putida*, the genes *putA* and *putP* are adjacent and divergently transcribed. The *put* genes are regulated at the transcriptional level with proline acting as an inducer. In this context, PutA acts as a specific repressor of *putA* and *putP* expression, thus controlling the expression levels of the pathway, as described for enterobacteria [119]. Although no differences in PDH activity in a *cbrB* background have been detected in *P. aeruginosa* when using proline as carbon source [25], the expression of the *put* promoters in *P. putida* in the presence of proline requires an as-yet-unidentified positive regulatory protein, whose expression appears to be σ^N^-dependent because *put* genes were not expressed in an *rpoN* background [115]. Visual inspection of the promoter region of the *put* genes in *P. putida* reveals sequences that show a certain similarity with the CbrB binding consensus including a similar spacing between the individual subsites from each other (TcTTAC-N_13_-GcAACc-N_22_-GTgAaA). Whether CbrB is the transcriptional activator of these genes must be experimentally verified.

### 5.4. Leucine Catabolism

Pseudomonads can use branched amino acids such as leucine, valine, and isoleucine as less-preferred carbon sources. In *P. aeruginosa*, leucine is metabolized by enzymes encoded in the *liu* gene cluster [120,121]. The *liu* genes are arranged in two transcriptional units, *liuRABCDE* and *liuBCDE*, also harboring the gene encoding the LiuR regulator. In the presence of leucine, LiuD is strongly expressed but repressed in the presence of glucose or succinate. The pathway is also under the translational control of Crc/Hfq [120]. In the presence of preferred carbon sources, transcription is initiated from promoter P1*_liuR_* but data indicate that transcription may be arrested at either of the two terminators located downstream *liuR*. Expression of LiuR under this condition assures low expression levels of the *liu* gene cluster. The activity of LiuR is antagonized by a putative protein that responds to the presence of leucine or citronellol favoring transcription from the P2*_liuB_* promoter [121]. Under this condition (non-preferred carbon source), expression of the sRNA CrcZ antagonizes the repressing effect of the Crc/Hfq complex on the translation of the *liu* mRNAs, allowing the catabolism of leucine and acyclic terpenes. On the other hand, the authors show that the CbrB response regulator was required for expression of LiuD in all the carbon sources tested [121]. The *liuRABCDE* transcript may be initiated by two putative promoters, one of them being dependent on σ^N^, which is activated under carbon deprivation [121]. It is likely that this promoter is directly activated by CbrB under such conditions, but the molecular mechanism of the promoter activation remains to be established. This would imply that the *liu* cluster is regulated at the transcriptional and translational levels by CbrAB and Crc/Hfq, respectively, while LiuR would act as a transcriptional repressor at another regulatory level.

## 6. Influence of the CbrAB-Mediated Carbon Control on Other Regulatory Systems

### 6.1. ECF Sigma Factors

The regulatory hierarchy of carbon source assimilation in *Pseudomonas* comprises a complex network that extends beyond CbrAB. In this network, we find control systems at the transcriptional level, including ECF sigma factors, as well as post-transcriptional regulatory elements such as the Crc/Hfq complex. ECF-type sigma factors often act as alternative transcriptional regulators to the major factor σ^70^ in response to specific environmental signals [122]. The genus *Pseudomonas* contains a large number of these factors encoded in their genomes, which represents an advantageous regulatory mechanism for generating a rapid response to environmental fluctuations [123,124]. Typically, the σ^ECF^ factor is sequestered by its anti-sigma factor (anti-σ) partners, many of which are membrane anchored, in the absence of the inducing stimulus [125,126,127,128]. However, there are a few cases in which protein–protein interactions between σ/anti-σ factors have been characterized at the molecular level [129].

Among the σ^ECF^, the SigX factor (σ^SigX^) in *P. aeruginosa* is involved in maintaining membrane fluidity and providing increased resistance to the action of different antimicrobial peptides [130]. A σ^SigX^ mutant showed impaired cell morphology, fluidity and alterations in membrane fatty acid composition [36,131,132]. In addition, under osmotic stress conditions, SigX activates the transcription of *oprF*, encoding a major outer membrane porin in *Pseudomonas* [133,134]. The genomic organization of *sigX* (*PP2088* in *P. putida* KT2440 and *PA1776* in *P. aeruginosa* PAO1) is highly conserved in all sequenced genomes of the *Pseudomonas* genus [135]. Upstream *sigX* is the *cfrX-cmpX* operon containing the *cmpX* gene encoding a conserved cytoplasmic membrane protein with a MscS domain typical for mechanosensitive channels that provide protection against hypoosmotic shock [136]. To date, σ^SigX^ is considered an orphan σ^ECF^ because a possible anti-σ factor has not been located in the same operon as is often the case with other σ^ECF^. Nevertheless, CfrX has been postulated to act as its anti-σ pair [137]. In addition, σ^SigX^ regulates its own expression and its transcription could be modulated by an element located upstream the *cmpX-sigX* intergenic region [138,139,140]. The main consequence of the absence of SigX in *Pseudomonas* is a reduction in membrane fluidity, leading to an alteration in the anchoring of membrane transporters, and as a consequence, in nutrient uptake. All this leads to a deregulation of genes involved in energy and nutritional metabolism, which corresponds to the most drastic effect reported for a *sigX* mutant [36]. Indeed, many of the genes of the SigX regulon are also found in the Anr, CbrB, GacA, LasR, RhlR and Crc regulons [36,141].

In *P. aeruginosa*, the Δ*sigX* mutant revealed some alterations in the regulatory pathways involving the Crc/Hfq complex as well as CbrAB [36,142]. Comparison of expression gene profiles in rich medium between a *crc* and *sigX* mutant showed that the expression of 48 genes under control of Crc was compromised after deletion of the *sigX* locus. Some of them were genes involved in the transport and metabolism of amino acids (*bkdA1A2B*, *hutUHIG*, *liuABCD* and *phhAB* operons), glutamate or derivatives (porins OpdP, OpdQ), aspartate (*aat* genes), branched-chain (*braC*) and aromatic (*aroP2*) amino acids, organic acids (OpdH porin) as well as *dctA* that encodes a succinate and fumarate transporter, that is tightly regulated by Crc/Hfq in *P. aeruginosa* [36]. Finally, the expression of *amiE*, which encodes a short-chain aliphatic amidase, is also reduced in a *sigX* background, at a transcriptional and translational level in PAO1. This gene is used as a reporter for CCR, and confirms that the Crc-based regulation is strongly reduced in a *sigX* mutant when grown in LB. On the other hand, the CbrAB system was found to be strongly activated in *P. aeruginosa,* even at conditions of catabolite repression. This was shown as an increase in the expression of CrcZ sRNA in a Δ*sigX* background, compared to the wild type, which in turn would sequester the translational regulators Hfq and Crc, thus reducing CCR [36]. This expression pattern is also observed in a *sigX* background in *P. putida* KT2440 although it is somewhat less evident (our unpublished data). Altogether, these data suggest that SigX is indirectly involved in CCR, either through its effects on membrane integrity and fluidity, or by mediating a regulatory effect on some specific targets controlled by CbrB. To validate this hypothesis, the RNA expression levels were determined under conditions of CbrAB induction, i.e., under carbon-limiting conditions (our unpublished results) showing that CrcZ levels in non-referenced C sources were not altered in a *sigX* background compared to the of *P. putida* wild-type strain, suggesting an indirect effect of SigX on the sRNA levels and independent of a CCR event.

On the other hand, the CrcZ levels in a *crc* background are drastically reduced (10-fold) compared to a wild-type strain in LB medium [72], thus showing an antagonistic effect to that evidenced in the *sigX* mutant. However, the double deletion of *crc* and *sigX* shows a dominant epistatic effect of Crc over the possible regulation exerted by SigX, as CrcZ levels in the double mutant were equivalent to those of the *crc* background (unpublished data). These results, together with the existence of an overlap between Crc and SigX regulons [36,141], would explain a double regulation of CrcZ, in which Crc would be above SigX in the regulatory hierarchy.

### 6.2. The TCS GacS-GacA

GacSA is a widely distributed and well-studied TCS in γ-proteobacteria. It represents a paradigm of signal transduction systems having pleiotropic effects as a global regulator [143]. Of the more than 60 TCS encoded in the *Pseudomonas* genomes, the analogy in the regulatory mechanism of the pathways CbrAB and GacSA is evident. The mechanisms involve the participation of regulatory RNAs that bind regulatory proteins acting at the post-transcriptional level. CbrAB activates the expression of the sRNAs CrcZ and CrcY, which bind the Crc/Hfq complex [74,75,78,144], whereas GacSA modulates the expression of the sRNAs RsmZ, RsmY and RsmX, which bind and inactivate the proteins RsmA and RsmZ [145,146,147,148,149].

In brief, the sensor kinases CbrA and GacS phosphorylate their cognate response regulators, CbrB and GacA, after sensing as-yet-unidentified environmental signals [143]. However, the domain architecture of GacS is distinct from CbrA because it also contains REC and Hpt (*Histidine phosphotransfer*) domains, being considered an unorthodox kinase. Furthermore, the signaling activity of GacS is regulated antagonistically by kinase–kinase interactions between RetS, LadS and PA1611 [150,151,152] (Figure 5). LadS upregulates GacS after detection of Ca^2+^ abundance while RetS downregulates GacS and blocks GacS autophosphorylation [153,154]. In the course of this phosphorelay process, PA1611 also inhibits the activation of GacS through its interaction with RetS [155,156].

From the discovery of the TCS GacSA, its involvement in numerous biological processes has been described due to the extent and variety of its regulatory output [157,158,159]. In *Pseudomonas* sp., GacSA significantly influences bacterial physiology by regulating the transition between surface-associated and planktonic lifestyles, essential in the transformation from acute to chronic infections. During this transition, GacSA intervenes through the Rsm proteins in the switching between type-3 and type-6 secretion system (T3SS, T6SS) [160], biofilm formation, polysaccharide synthesis, iron uptake [161] as well as synthesis of quorum-sensing signals [158,162,163]. Moreover, in *A. vinelandii*, it participates in the synthesis of alkylresorcinols that are related to the encystment process [164], two polymers of biotechnological interest, poly-β-hydroxybutyrate and alginate [165,166], and also activates motility and flagella synthesis [167].

On the other hand, CbrAB controls several processes that overlap some of the GacSA regulatory output in *Pseudomonas* and *Azotobacter*, such as carbon metabolism [25,32,95,168], motility and flagella [33,90,169,170] and synthesis of compounds and polysaccharides [99,171]. Beyond its relevant role in catabolic repression, Crc together with Hfq also modulates processes such as biofilm formation [172,173], production of pyocyanin [174,175] and rhamnolipids [176], quorum sensing (QS) [177], secretion of proteinaceous virulence factors [178] and antibiotic resistance [179]. The dual involvement of these regulatory systems in diverse processes of bacterial physiology has sparked interest in the search for common links that demonstrate their interaction. In fact, in many of the studies performed in *Pseudomonas* sp., mutants in CbrAB or GacSA shared phenotypic characteristics that suggested cross-talk between both signaling pathways [180]. In addition, the sequestration of RsmA/N by RsmZY caused changes in the genetic expression pattern that are similar to those that occur with Hfq and CrcZY [144,146] (Figure 5).

Although the issue has not been explored in much detail, Romero et al. have evidenced that both RsmN and RsmA exhibited specific but distinct CrcZ-interacting motifs (RsmN binds to the 5′ region while RsmA in the 3′ region of CrcZ transcript) [181]. Furthermore, it has been shown that Hfq is able to bind and stabilize the sRNA RsmY [182]. Another clear example of the interplay between both systems is the dual regulation of *lipA* gene expression by CbrB and GacA, whose transcription has been shown to be positively controlled by both of them [86,183] (Figure 5). Zha et al. also reported that both RsmA and RsmE, a RsmA-CrsA family of transcriptional regulators in *P. fluorescens* Pf-5, are able to control its translation. While RsmA inhibits *lipA* transcription through an unidentified mechanism, but enhances its translation, RsmE represses its translation [183]. All these phenomena establish a stellar map of related elements that could form a multikinase network of cooperation between diverse systems to achieve a balanced physiology.

## 7. Integrated Control of the Virulence in *Pseudomonas*

Bacterial virulence involves different mechanisms of a diverse nature that promote cytotoxicity towards eukaryotic host cells. During pathogenesis, bacteria develop a wide repertoire of fascinating mechanisms to colonize new environments. These involve changes in bacterial motility, the synthesis of virulence factors and their translocation into the host cell, as well as the development of protective mechanisms to avoid the host immune response [184,185,186].

The World Health Organization has established a catalogue of multiresistant pathogenic micro-organisms in order to prioritize efforts in the search for new antimicrobial agents. *P. aeruginosa* is included among the critical priority pathogens, denoted ESKAPE (for *Enterococcus faecium*, *Staphylococcus aureus*, *Klebsiella pneumoniae*, *Acinetobacter baumannii*, *P. aeruginosa* and *Enterobacter* spp.) [187,188]. Amongst the *Pseudomonas* pathogenic strains, PAO1 and PA14 of *P. aeruginosa* are clinical isolates from cystic fibrosis (CF) patients, which have adapted to hostile environments such as human lungs or have even evolved to acquire nutrients from higher organisms [189,190,191]. Other pathogens such as *P. syringae* pv. *tomato* DC3000, a plant pathogen of *Arabidopsis thaliana* and *Solanum licopersicum*, have acquired other virulence mechanisms that allow them to reproduce at the expense of the infected plant [192,193]. Pathogenicity has been reported to be host specific [194], suggesting that certain *Pseudomonas* species exhibit a high potential for damage when suitable conditions are present [195,196]. Infections caused by these opportunistic pathogens often lead to clinical complications, especially in immunocompromised patients [197,198]. The pathogenicity associated with them lies frequently in the existence of pathogenicity islands in their genome as well as various associated virulence factors. The gaps in the knowledge of molecular processes, resulting in pathogenicity as well as the lacking annotation of more than 500 *P. aeruginosa* genes are among the obstacles that need to be overcome to accelerate the discovery of anti-virulence targets.

The principal virulence-driving processes and factors in *P. aeruginosa* and other pathogenic strains include biofilm formation, the motility provided by the flagellum or pili, the T3SS proteins (*exo*, *hrp*, *hrc*) and the biogenesis of effectors such as pyocyanin (*phz*), rhamnolipids (*rhl*), proteases and elastases (*las*). In fact, *P. aeruginosa* is the only microorganism known to date to synthetize pyocyanin, a redox-active phenazine capable of accelerating neutrophil apoptosis in vitro [199]. In this contest, it also secretes rhamnolipids, a type of hemolysin that provides an advantage in modulating the host immune response [200,201]. The literature provides evidence that TCSs such as CbrAB or GacSA, in concert with other HKs, are involved to effectively regulate these virulence-related processes [159,163] (Figure 6).

One of the main reasons why CF infections become chronic, as well as mucoid conversion, is the formation of biofilm, which represents a defensive structure against phagocytosis and permits the development of antibiotics resistance, during chronic infections [202,203,204,205]. TCSs participate in the control of both processes. Whereas the MucA/AlgU anti-σ/σ pair and the AlgZR TCS control the overproduction of the alginate polysaccharide, the GacSA and RsmA/RsmZY systems regulate the synthesis of the Psl and Pel polysaccharides that are required for biofilm formation [206,207,208] (Figure 6). The Pel and Psl exopolysaccharides (EPS) are responsible for forming rugose small colony variants (RSCV) and are one of the main causes of antibiotics resistance in *Pseudomonas* [209,210,211]. On the other hand, motility plays a relevant role in the virulence of *Pseudomonas,* as it is the main way to colonize diverse ecological niches, from open to restricted spaces. Bacteria react to stimulus gradients and show different motility patterns according to the viscosity of the medium: swimming or swarming for semi-solid surfaces and twitching for solid surfaces [212,213,214]. Although flagella appear to be the most important means of propelling [215], swarming requires type IV pili and rhamnolipids production while twitching is only based on type IV pili action [216,217]. The synthesis and action of the flagellar apparatus represent a very important metabolic burden to the cell which is compensated by a larger number of benefits. The flagellar gene arrangement and regulation in Pseudomonads stands out for its complexity, since it occurs by a three- or four-tier cascade mechanism (classes I–IV). At the top of this transcriptional hierarchy is FleQ, a σ^70^-dependent transcriptional regulator like CbrB, which triggers the sequential cascade, leading to the activation of gene clusters that structural flagellar components, as well as diverse proteins of the chemotaxis machinery and accessory regulatory proteins [218,219].

Transposon mutagenesis approaches in *P. aeruginosa* PA14 have led to the identification of a *cbrA* mutant that showed increased biofilm formation and no swarming motility suggesting a reverse regulation of these two processes [90]. Furthermore, a *cbrB* mutant in *P. putida* also showed altered swimming motility, and its transcriptome analysis revealed altered expression of genes encoding functions potentially involved in biofilm formation, such as signal transduction proteins containing GGDEF domains, exopolysaccharide biosynthesis or transport as well as the synthesis of the adhesin protein LapF [33]. In the fungal phytopathogen *P. aeruginosa* PGPR2, a highly efficient root colonizer and biocontrol strain against *Macrophomina phaseolina*, an insertion mutant in *cbrA* resulted in hampered corn colonization, which was associated with a significant increase in biofilm formation, as well as the deregulation of genes involved in alginate production (*algD* and *algU*), EPS synthesis (*pelB* and *pslA*) and motility (*fliC* and *flhF* coding flagellin subunit and a flagellar biosynthesis regulator, respectively) [220,221].

CbrA, through CbrB, was found to be involved in *P. aeuginosa* PAO1 cytotoxicity towards HBE (*Human Bronchial Epithelial*) cells [90,170]. CbrA function also results in a resistance to clearance by phagocytes, the first cellular barrier of the immune system against pathogens, since a *cbrA* mutant was 13-fold more resistance to be killed by neutrophils than wild-type strain. A similar effect was obtained using the non-mammalian host system model *Dictyostelium discoideum*. Furthermore, a comparison of gene expression profiles between a *cbrA* defective mutant and the wild-type during *D. discoideum* infection, allowed to identify 286 genes that were specifically regulated by CbrA [170].

Crc is a master regulator that has several pleiotropic effects in *Pseudomonadaceae* including virulence, since it controls virulence determinants production as well as T3SS. In *P. aeruginosa*, Crc has been shown to modulate the expression of the virulence factors Hcp1 (a putative type VI secreted effector), HcnB (a hydrogen cyanide synthase), and Azu (the azurin precursor involved in cytotoxicity) [179]. The authors also suggested cross-talk between quorum-sensing and metabolism through the production of hydrogen cyanide, an inhibitor of the ciliary beat frequency of respiratory cells [222]. Crc also conditions antibiotic susceptibility, expression of T3SS- and QS-regulated virulence factors, such as pyocyanin [174] and potentially other virulence functions [179]. In *P. syringae* pv. *tomato* DC3000, the inactivation of *crc* locus resulted in altered swarming motility and virulence patterns as well as high susceptibility to hydrogen peroxide [172]. It has been speculated that the reduced virulence phenotype might be due to the role of Crc in regulating c-di-GMP levels and even in the carbon catabolite repression. Several sugars and organic acids are abundant in apoplast extracts and their assimilation permits efficient plant colonization.

On the other hand, the T3SS is the primary virulence mechanism to inject virulence factors into the cell, resulting in an inhibition of phagocytosis and tissue necrosis in turn. In a number of cases, T3SS promotes the mutualism relationship between the Plant Growth-Promoting Rhizobacteria (PGPR) and the plant host [223,224,225]. A T3SS consists of four principal components: the basal body that anchors the complex to the membranes, the export gate to obtain energy for the secretion process, the needle filament and the translocation pore through which proteins and effectors are translocated [226]. The mechanism of invasion depends on the nature of the host cell which in turn influences T3SS architecture. An example is the difference in the length of the extracellular needle between plant and animal pathogens, which conditions the transport of exotoxins into the host cell [227]. In many animal pathogens, ExsA is the master regulator of the T3SS, inducing the synthesis of the T3SS components and various virulence effectors, such as ExoT, ExoU, ExoS and ExoY, which are translocated into the host cell [186]. The expression of *exsA* is upon positive control of Vfr RR, which requires cAMP for its binding to the *exsA* promoter (Figure 5). Under specific conditions (Ca^2+^ depletion, direct cell contact or presence of serum), ExsA is released from the heterodimer formed with ExsD and promotes the expression of its regulon [186].

T3SS-related genes in plant pathogens are named differently: *hrp* (hypersensitive response and pathogenecity) and *hrc* (hypersensitive response and conserved) genes that along with other effectors are located on a pathogenicity island [228,229]. Despite these differences, Shao et al. have established cross-talk of as-yet-unknown pathways by compiling and combining data from RNA-seq and ChIP-seq of 16 virulence-associated regulators in *P. syringae*. Particularly, CbrAB was identified to represses the master regulators HrpR, HrpL and HrpS and T3SS effectors (*hrpK1*, *hrpA2* and *hrpZ1*) in carbon excess conditions and to activate them under carbon limitation [230]. Next to the control exerted by CbrAB, GacSA mediates in *P. syringae* analogous regulatory circuit, through the control of the *hrpRS* operon together with the CvsRS and RhpRS TCSs [231]. There is evidence that CbrAB and Crc/Hfq are involved in the regulation of the T3SS and its effectors [90,232] (Figure 5). Dong et al. suggested that Crc indirectly represses T3SS expression through changes in the concentration of intracellular metabolites such as histidine, whose transport and metabolism prevent the expression of T3SS-related genes [233].

Overall, the indirect effects of the CbrAB system detected at different stages of pathogenesis in *Pseudomonas* sp. seem to converge with those of the GacSA system. In addition, RmsA also participates as a global regulator, controlling T3SS genes such as *exsC* and *exsD* encoding regulatory proteins and *popB*, a translocalizing protein (Figure 5). However, these three genes manifested an interesting opposite phenotype in *cbrAB* mutants [221]. Indeed, several studies have rigorously compiled all TCSs contributing to virulence in *P. aeruginosa*, highlighting the involvement of the GacSA network for its decisive role in the conversion from acute to chronic infection [159,163,234], noting that the key to pathogenesis is the efficient integration of signal inputs into an extensive multikinase network in which the second messengers cAMP and c-di-GMP act as exchange currencies.

## 8. Conclusions and Final Remarks

Although CbrAB has been studied to some extent, much of its regulatory mechanism and it related phenotypic consequences remain to be identified. A central question about CbrA relates to its non-canonical topology, which comprises a transporter and a sensor kinase unit. The advantages of coupling substrate uptake with signaling, particularly in bacteria harboring a large number of homologous transporters of the same substrate, appears to be of evolutionary significance. In general, regulation of gene expression can be mediated by single-protein regulators that represent a much lower metabolic burden to the cell than TCS that are composed of multiple proteins that require ATP to function. Thus, the potential advantages of CbrAB-mediated regulation compare to those that could be exerted by a single-protein regulator. It may be hypothesized that this is related to the complexity to the CbrA topology that may allow different signals to be potentially integrated into its SLC5, STAC and PAS domains. Regulatory processes that rely on the recognition of multiple different signals allow fine-tuning of the regulatory output, which may also be a major force that has driven the evolution of CbrA. However, the precise identification of signals integrated by CbrA remains an open question and in the future will allow for a better understanding of the corresponding regulatory circuitry.

## Figures and Tables

**Figure 1 genes-13-00375-f001:**
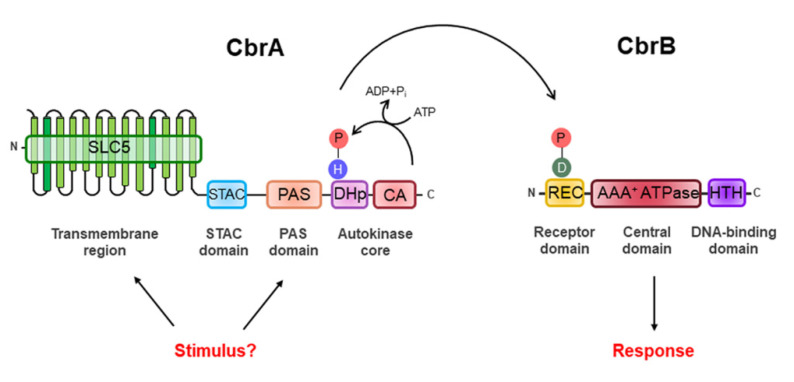
Schematic representation of the TCS CbrAB system in *Pseudomonadaceae*. The histidine kinase CbrA bears two domains susceptible to detect an environmental signal (SLC5 domain and PAS domain). Stimulus detection modulates autophosphorylation at a highly conserved histidine residue located in the dimerization and phosphotransfer domain (DHp) in an ATP-dependent reaction catalyzed by the catalytic domain (CA). Immediately, CbrA transfers the phosphoryl group to a conserved aspartic residue in the REC domain of CbrB. Phosphorylation of CbrB promotes its oligomerization and binding to the promoter regions of its regulon through the DNA-binding domain.

**Figure 2 genes-13-00375-f002:**
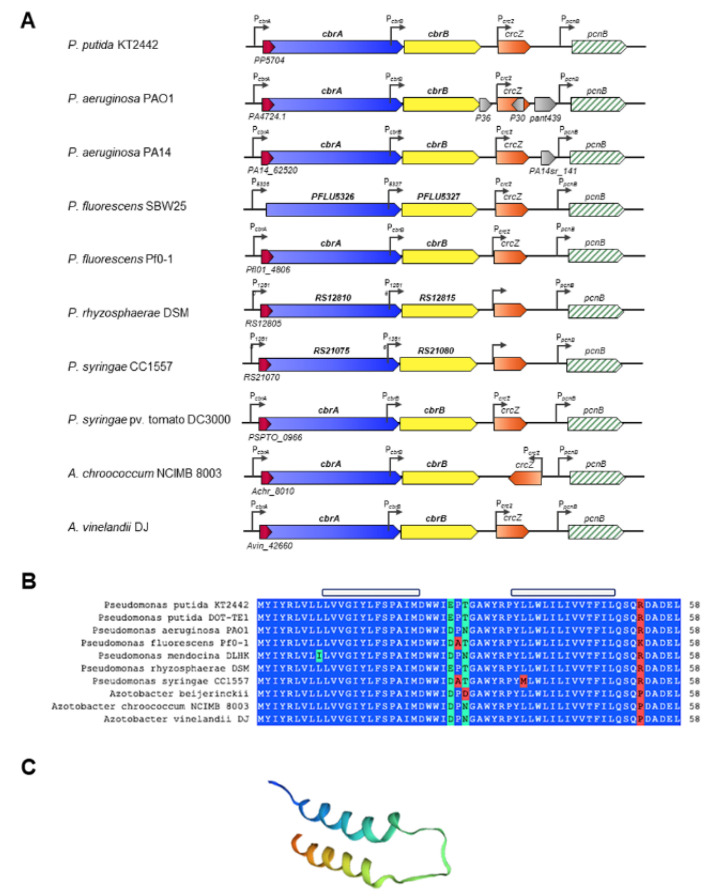
(**A**) Genomic organization of the *cbrA–cbrB–crcZ* locus in the family *Pseudomonadaceae*. Graphical representation of the gene organization of *cbrX* (burgundy), *cbrA* (blue), *cbrB* (yellow) and *crcZ* (orange) in *P. putida* KT2442, *P. aeruginosa* PAO1, *P. aeruginosa* PA14, *P. fluorescens* SBW25, *P. fluorescens* Pf0-1, *P. rhyzosphaerae* DSM 16299, *P. syringae* CC1557, *P. syringae* pv. *tomato* DC3000, *A. chroococcum* NCIMB 8003 and *A. vinelandii* DJ. Promoter regions are shown as arrows, other putative piRNAs in grey and adjacent genes in green. (**B**) Sequence alignment of CbrX in different strains of the *Pseudomonadaceae* family. Identical amino acids are indicated in blue, amino acids with similar physicochemical characteristics in green and non-conserved amino acids in red. Sequences corresponding to the α-helices depicted in panel B are indicated at the top of the sequence. (**C**) Secondary structure of CbrX predicted by homology modelling of protein structures by SWISS-MODEL (ExPASy).

**Figure 3 genes-13-00375-f003:**
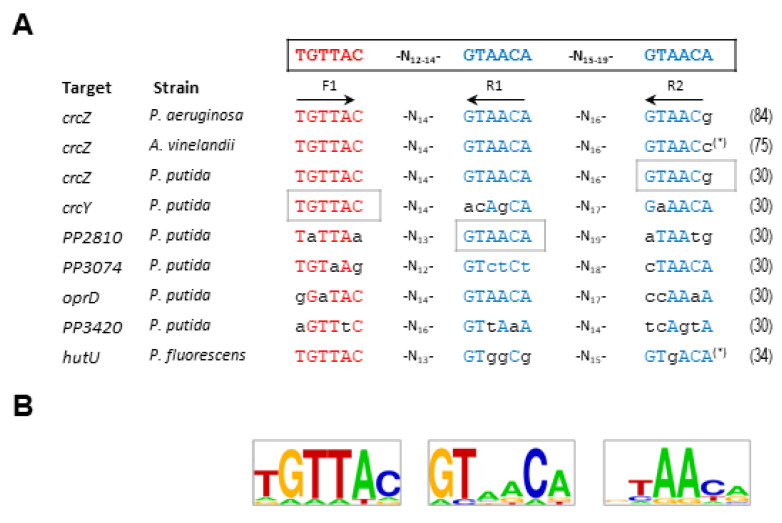
(**A**) Sequence alignment of the putative CbrB binding subsites for those targets that have been experimentally shown by mutagenesis to be CbrAB controlled in different organisms. The consensus sequence and orientation are shown at the top. Colored sequences in uppercase coincide with the proposed consensus. In grey boxes are the most relevant subsites for transcriptional activation for *crcZ*, *crcY* and *PP2810* in *P. putida*. The asterisk (*) shows that the consensus sequence established by the authors has been re-evaluated in this work and the third binding subsite is proposed at the indicated distance. (**B**) LOGO sequence obtained from the sequences experimentally assayed as CbrB binding sites.

**Figure 4 genes-13-00375-f004:**
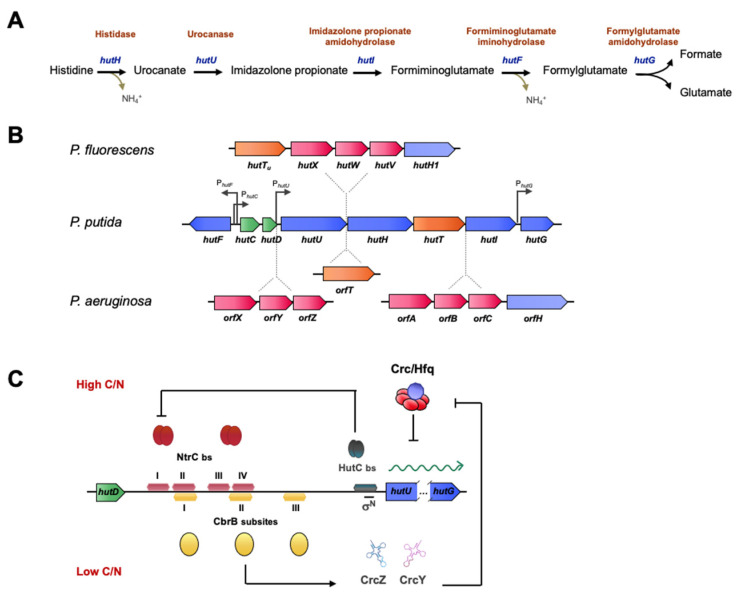
(**A**) Conserved histidine catabolic pathway in *Pseudomonas*, that yields 2 moles of ammonia, 1 mole of glutamate, and 1 mole of formate per mole of histidine. Genes coding for each activity are denoted in blue and the enzymes in brown. (**B**) Genetic organization of the *hut* operons in three *Pseudomonas* species (*P. fluorescens*, *P. putida* and *P. aeruginosa*). Genes with the same colors are homologous (catabolic enzymes in blue; histidine transporters in orange; ABC-type transporters in pink; other *hut* genes in green). (**C**) Carbon and nitrogen regulation of the *hut* genes by CbrAB, NtrC, Crc/Hfq and HutC regulators. In high-carbon-availability conditions, NtrC activates the catabolism of histidine, and HutC slows down the production of ammonia, which inhibits the process. Crc/Hfq represses the translation of the *hut* mRNAs in these conditions. At low C concentrations, CbrB is active and binds the promoter region of *hutU*. The sRNAs CrcZ and CrcY are highly activated by CbrB and sequester the Crc/Hfq complex, which is inactivated, thus promoting the translation of *hut* transcripts.

**Figure 5 genes-13-00375-f005:**
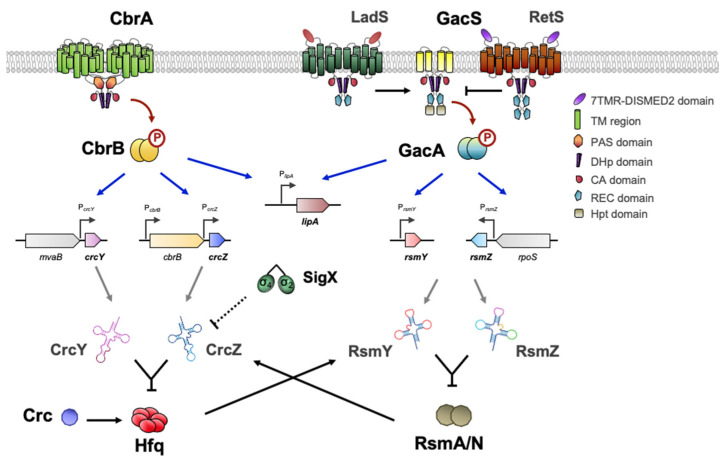
Representation of the parallelism and cross-talk of the CbrAB and GacSA regulatory systems in *Pseudomonas*. Under induction conditions, the sensor kinases CbrA and GacS are activated by autophosphorylation in an ATP-dependent reaction, and subsequently transfer the phosphoryl group to their response regulators, CbrB and GacA, respectively. Phosphorylation of CbrB and GacA leads to their oligomerization and binding to promoter regions of sRNAs genes (*crcZ* and *crcY* for CbrB; *rsmZ* and *rsmY* for GacA) and other target genes such as *lipA*. Sufficient amounts of the sRNAs promote sequestration of Hfq and RsmA or RsmN, by direct binding to specific domains of the proteins. Each of these regulatory pathways also converge in the protein–protein interaction between Hfq and RsmY, and RsmN and RsmA with CrcZ. On the other hand, the sigma factor SigX has a negative effect on the CrcZ levels. Blue and black arrows represent transcriptional and post-translational regulation, respectively. Dotted lines indicate an indirect regulation. The domains that make up the architecture of the sensor kinases are shown in the legend.

**Figure 6 genes-13-00375-f006:**
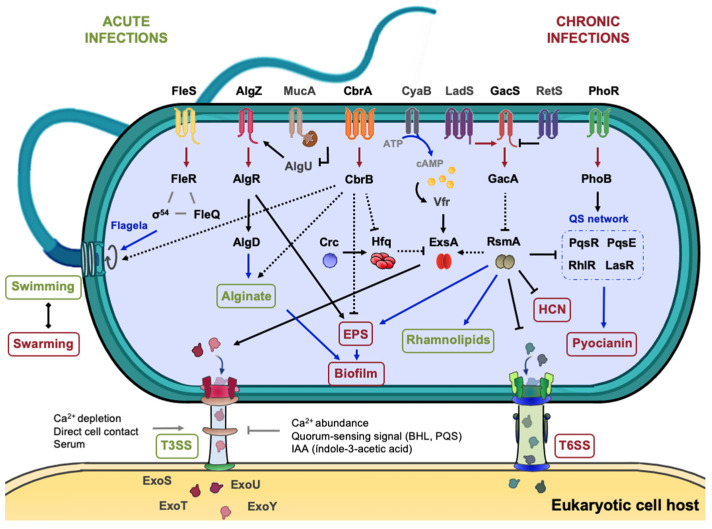
Interaction of multiple pathways involved in mediating *P. aeruginosa* virulence. FleS, AlgZ, CbrA, GacS and PhoR sensor kinases activate, under induction conditions, their cognate RRs, FleR, AlgR, CbrB, GacA and PhoB, respectively, triggering the activation of acute (boxed in green) or chronic (boxed in burgundy) phenotypes. Biogenesis of pyocyanin is tightly regulated by GacSA-RsmA as well as by the quorum-sensing network. The QS network involves the participation of RRs such as PqsR, PqsE, RhlR and LasR, which are downregulated by RsmA and upregulated by the TCS PhoR/B. Flagellum biogenesis depends on the induction of the TCS FleS/R, where FleR acts as RR together with FleQ. Binding of both proteins to σ^N^-dependent promoters activates flagellum formation in sequential cascades. The antagonistic effect of Hfq and RsmA on the master regulator of the T3SS, ExsA, is also shown: Vfr promotes ExsA expression in a cAMP-dependent reaction. Red, blue and black arrows indicate phosphotransfer reactions, induction/formation and activation/repression regulation, respectively, with dotted lines indicating an indirect regulation. EPS: exopolyssacharides; HCN: hydrogen cyanide; T3SS: type-3 secretion system; T6SS: type-6 secretion system; BHL: N-butyryl homoserine lactone; PQS: *Pseudomonas* quinolone signal.

## Data Availability

Not applicable.

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
