# Peer review of "The Regulatory Hierarchy Following Signal Integration by the CbrAB Two-Component System: Diversity of Responses and Functions"

_genes, 2022, doi:10.3390/genes13020375_

Round 1

Reviewer 1 Report

The authors have been doing active research in this specific area, and the manuscript is well written in general. Few comments are listed below for consideration during revision.

  • Figure 3, Figure 4, lines 388-411: The precise mechanisms of CbrB action are not clear. It would thus be more helpful to focus on the empirical evidence available in literature rather than the previous implications based on in silico sequence analysis.
  • (1) Recommend deleting Figure 3B. The promoter prediction made by Abdou et al (2011) was informative (and helpful), but no researchers would take these in silico predictions very seriously without experimental verification. Therefore, it is not necessary to highlight these contradicting (or incorrect) predictions in a figure, which will potentially mislead some readers.
  • (2) Regarding PhutU of P. fluorescens SBW25, Naren and Zhang (2021) have recently shown that introduction of the “GTAACA” site (the R1 site here) into PhutU enhanced protein/DNA interactions in EMSA. More importantly, we consider the absence of this R1 site is functionally relevant, and it is evolved as a mechanism to reduce CbrB-mediated promoter activities. This is important for CbrAB to balance its role in controlling hut gene expression at the levels of both transcription and translation. Of course, we can argue that there is a weak CbrB binding site (R1) over there. Therefore, it is important to present the facts clearly and avoid unnecessary confusions in literature. Consensus logos should be provided for an appropriate sequence analysis.
  • (3) I doubt that “none of the three is essential for transcriptional activation” (line 410), so please cite any supporting evidence. Our genetic data and EMSA showed that the first site (F1 here) is essential for CbrB function (Fig. 1 and 2, Naren and Zhang 2021). There are other studies also showing that the first F1 site plays the predominant role for CbrB binding.
  • Line 841. It sounds incorrect to say that CbrB was not involved in PAO1 cytotoxicity. Yeung et al revealed no phenotypic change of cytotoxicity with the cbrB mutant. This is the fact, but it doesn’t exclude the possibility that CbrB is involved in determining cytotoxicity (and I doubt Yeung et al have said that in their papers). In general, sensor kinase and response regulator mutants can have different phenotypes.
  • Line 71. As far as this reviewer remember, the role of CbrAB in maintaining C/N balance has been tested by specifically designed experiments in aeruginosa by Nishijyo et al (2001) (last section of Results) and Zhang and Rainey (2008) (Figure 5). Please note the differences between C/N control and CCR.
  • Line 479, ref. 105, I don’t think in this cited work that CbrAB has been shown to “directly” activate the aot Please check.
  • Line 505, hutG
  • Line 497-519: Re. references related to hut genes in P. fluorescens SVBW25, the hut catabolic genes, including hutH1 and hutH2, were characterized in Zhang and Rainey 2007 (doi: 10.1534/genetics.107.075713), transporter genes in Zhang et al. (2012, ref. no. 113).
  • Line 512: hut promoters or hutC promoter? Additionally, it should be “absence”, not “presence”.
  • Line 516, please don’t use the word “redundant”, as only one of them encodes a functional histidase.
  • Line 521, pseudomonads, not “Pseudomonads”.
  • Finally, I hope authors can make some comments on the multiple copies of ncRNAs, as they are crucial for understanding CbrAB function across different species of Pseudomonas. In a previous work we addressed this phenomenon using the method of mathematical modelling (Liu et el. 2017, doi: 10.1111/mmi.13720). Results showed that the multiple copies of ncRNAs are not functionally redundant, and they must possess distinct biological functions for their stable maintenance in the genomes of certain Pseudomonas species. I hope this review can help encourage further molecular studies to search for different functions of CrcY, CrcZ and CrcX.

Xue-Xian Zhang

https://sites.google.com/view/xxzhang/home

Author Response

see doc attached

Reviewer 2 Report

The review under the title"The regulatory hierarchy following signal integration by the 2 CbrAB two-component system: diversity of responses and functions" is well written and the topic is important in the field although there are several papers studied this issue, the current review has been presented the topic differently. Just I suggest the authors can include the phylogenetic tree for the sequence alignment of the putative CbrB. 

Author Response

see doc attached
